

# Effects of grazing intensity and the use of veterinary medical products on dung beetle biodiversity in the sub-mountainous landscape of Central Italy

Mattia Tonelli[1,2], José R. Verdú[2] and Mario E. Zunino[1,3]

[1] Department of Pure and Applied Science, University of Urbino, Urbino, Italy
[2] I.U.I. CIBIO, Universidad de Alicante, Alicante, Spain
[3] School of Biodiversity, Asti University Centre for Advanced Studies, Asti, Italy

Corresponding author
José R. Verdú, jr.verdu@ua.es

## ABSTRACT

Grazing extensification and intensification are among the main problems affecting European grasslands. We analyze the impact of grazing intensity (low and moderate) and the use of veterinary medical products (VMPs) on the dung beetle community in the province of Pesaro-Urbino (Italy). Grazing intensity is a key factor in explaining the diversity of dung beetles. In the case of the alpha diversity component, sites with a low level of grazing activity—related in a previous step to the subsequent abandonment of traditional farming—is characterized by a loss of species richness ($q = 0$) and a reduction in alpha diversity at the levels $q = 1$ and $q = 2$. In the case of beta diversity, sites with a different grazing intensity show remarkable differences in terms of the composition of their species assemblages. The use of VMPs is another important factor in explaining changes in dung beetle diversity. In sites with a traditional use of VMPs, a significant loss of species richness and biomass is observed, as is a notable effect on beta diversity. In addition, the absence of indicator species in sites with a historical use of VMPs corroborates the hypothesis that these substances have a ubiquitous effect on dung beetles. However, the interaction between grazing activity and VMPs when it comes to explaining changes in dung beetle diversity is less significant (or is not significant) than the main effects (each factor separately) for alpha diversity, biomass and species composition. This may be explained if we consider that both factors affect the various species differently. In other words, the reduction in dung availability affects several larger species more than it does very small species, although this does not imply that the former are more susceptible to injury caused by the ingestion of dung contaminated with VMPs. Finally, in order to prevent negative consequences for dung beetle diversity, we propose the maintenance of a moderate grazing intensity and the rational use of VMPs. It is our view that organic management can prevent excessive extensification while providing an economic stimulus to the sector. Simultaneously, it can also prevent the abuse of VMPs.

## INTRODUCTION

Land use changes play a pivotal role in the loss of biodiversity (*Sala et al., 2000*). In the Mediterranean Basin, starting about 10,000 years ago, the human population modified the landscape for agriculture and livestock grazing purposes (*Blondel, 2006*). Passing through the different stages that have characterized each era (*Vos & Meekes, 1999*), the Basin has developed a complex "cultural landscape" (cfr. *Farina, 2000*) that enables a large number of species to be maintained there (*Myers et al., 2000*). Semi-natural grasslands are one of the keystone habitats of this landscape. They were developed and managed by man (*Blondel et al., 2010*) using extensive livestock grazing that prevented the homogenization of the landscape (*Perevolotsky & Seligman, 1998*; *Diacon-Bolli et al., 2012*). This grazing also provides an energy input to the system through the cattle dung that was previously produced by wild herbivores.

In these semi-natural grasslands, dung beetles are among the most important groups within the dung fauna (*Hanski & Cambefort, 1991*). Their bionomics involves them, directly and indirectly, in various ecological processes such as: nutrient cycles, vegetation development, secondary seed dispersal, and parasite control (*Halffter & Matthews, 1999*; *Nichols et al., 2008*). Dung beetles fulfil all the characteristics of an ideal bioindicator taxon (*Spector, 2006*; *Halffter & Favila, 1993*), and have been used in a great number of studies on habitat disturbance or conversion (*Braga et al., 2013*; *Halffter & Arellano, 2002*; *McGeoch, Van Rensburg & Botes, 2002*); the natural environmental gradient (*Jay-Robert, Lobo & Lumaret, 1997*; *Romero-Alcaraz & Ávila, 2000*); and the vegetation and landscape structure (*Numa et al., 2009*; *Verdú, Numa & Hernández-Cuba, 2011*).

In the last few decades, extensive livestock management has undergone a rapid process of modification (*Stoate et al., 2009*). Italy has seen the progressive abandonment of traditional extensive grazing systems in favour of more intensive versions. Furthermore, from 1982 to 2010, Italian fields lost 20% of their heads of cattle (cows, sheep and horses), while the livestock of farms fell by about 71%. Nevertheless, the number of horses and sheep rose in the same period in valley areas (more than 13%) and hills (more than 12%), but fell by about 24% in mountain regions (*ISTAT, 2010*). Moreover, the number cow herds across the country has decreased by about 35% in the last 28 years, with 70% of cows concentrated in the north of Italy in 2010. Indeed, in this part of the country, the number of cow heads/farm increased from 48 to 64 between 2000 and 2010 (*ISTAT, 2010*; *Sturaro, Cassandro & Cozzi, 2012*). This has led to a situation where marginal areas are abandoned, but more productive locations can suffer from overgrazing. Another relevant factor related to intensification is the use/abuse of veterinary medical products (VMPs). These substances are widely utilized, with 194 tons of antiparasitic substances produced in the European Union in 2004 (*Kools, Moltmann & Knacker, 2008*). VMP molecules such as ivermectin are poorly metabolized by cattle (*McKellar & Gokbulut, 2012*) and are voided as unchanged residues in faeces (*Floate et al., 2005*; *Lumaret et al., 1993*). These residues have been demonstrated to have negative sub-lethal effects and ultimate lethal consequences on non-target dung fauna and, particularly, dung beetles (*Verdú et al., 2015*; *Wardhaugh, Longstaff & Morton, 2001*).

These three factors, i.e., grazing abandonment and intensification and VMP use, have been demonstrated to have negative effects on dung beetle biodiversity. Some studies have focused on the effects on dung beetles of grazing abandonment (*Jay-Robert et al., 2008*; *Verdú, Crespo & Galante, 2000*; *Carpaneto, Mazziotta & Piattella, 2005*), overgrazing (*Negro, Rolando & Palestrini, 2011*) and VMP use (for a review see: *Beynon, 2012*; *Lumaret & Errouissi, 2002*; *Wall & Beynon, 2012*; *Jacobs & Scholtz, 2015*).

When it comes to the impact of VMPs on dung beetles, however, the majority of research has been carried out in the laboratory, with the focus on the effects on a single or just a few species (*Verdú et al., 2015*; *Cruz-Rosales et al., 2012*; *Hempel et al., 2006*; *Wardhaugh & Rodriguez-Menendez, 1988*). Nevertheless, it is important to evaluate the impact of different grazing intensities in order to determine the optimum level for dung beetle conservation. This step is necessary because, increasingly, grazing activities are not being completely abandoned, but are instead suffering an ongoing process of extensification (*sensu* EUROSTAT: http://ec.europa.eu/eurostat/statistics-explained/index.php/Glossary:Extensification). Furthermore, pollutants (i.e., VMPs) may interact with "natural stressors" (i.e., the quantity of the trophic resource), producing synergistic or antagonistic effects (*Folt et al., 1999*; *Laskowski et al., 2010*). To our knowledge, no studies have evaluated the potential impact of the possible interaction of these two factors on dung beetle diversity.

The aim of this study was to analyze the effects of grazing intensity and the use of VMPs on dung beetle diversity in the sub-mountainous landscape of Central Italy. Comparing areas with different grazing intensities (low and moderate) and those with a historical use or non-use of VMPs (used as a proxy of intensification), we attempt to answer the following four questions: (A) What is the effect of grazing intensity and VMP use on: dung beetle alpha diversity at different Hill numbers or levels ($q = 0$, $q = 1$, and $q = 2$), abundance and biomass; (B) What is the possible interaction between these factors with respect to dung beetle diversity? (C) Are there any indicator species for a particular treatment; (D) What are the effects on the composition of dung beetle assemblages (beta diversity)? Our hypothesis is that a low level of grazing intensity and the use of VMPs have negative effects on dung beetle biodiversity, resulting in changes in alpha and beta diversity and biomass, and favouring the presence of some species that may act as indicators of a particular form of pasture management. Moreover, we hypothesize that the effects of low grazing intensity and VMP use are worse in combination than alone.

## MATERIALS AND METHODS

### Study area and experimental design

The study was carried in the sub-mountainous area of the Pesaro-Urbino province in the Marche region, Italy. The provincial climate falls into the temperate Köppen categories (Cfa and Cfb). The average annual temperature is around 12 °C, with a minimum average of around 3.5 °C in winter and a maximum average of 21 °C in summer. Average annual precipitation is around 930 mm, with two dry periods, one in summer and another in winter (www.pesarourbinometeo.it). The soil is calcareous.

To evaluate the effects of grazing intensity and VMP use, we designed a 2x2 full factorial design with three replications for each treatment. We identified different areas with: a

VMP-free, low grazing intensity; a VMP-free, moderate grazing intensity; a VMP-use, low grazing intensity; and a VMP-use, moderate grazing intensity.

(A) 'Low grazing, VMP-free' areas—LGECO—(Pietralata pastures; 43°39′33.64″N; 12°42′27.65″E). These secondary grasslands, located between 750 and 900 m a.s.l., are represented by the *Brizo mediae-Brometum erecti* and *Festuco circummediterraneae-Arrhenatheretum elatioris* associations. These grasslands are mainly used by horses that were abandoned and have reverted to a wild state. The grazing intensity of these pastures is around 0.7 units of livestock/ha. The most common wood species are: *Fraxinus ornus* L., *Ostrya carpinifolia* Scop., *Quercus ilex* L., *Quercus pubescens* Willd., *Acer opalus* (Miller), *Pinus nigra* J.F. Arnold, *Crataegus monogyna* Jacq., *Juniperus oxycedrus* L., *Lonicera etrusca* G. Santi, *Spartium junceum* L., and *Rosa canina* L.

(B) 'Moderate grazing, VMP-free' areas—MGECO—(Montebello pastures; 43°43′13.83″N; 12°45′19.98″E). These grasslands are located between 500 and 600 m a.s.l. within the Gino® Girolomoni Cooperativa Agricola. The pastures are used by cows according to organic farming rules with grazing rotation. The grazing intensity is about 1.5 units of livestock/ha. The herbaceous association falls within the *Brizo mediae-Brometum erecti* group. The spontaneous arboreal vegetation is prevalently comprised of *Quercus pubescens, Quercus cerris* L., *Quercus petraea* (Matt.) Liebl., *Carpinus betulus* L., *Ostrya carpinifolia, Fraxinus ornus* L., *Acer opalus, Quercus ilex, Sorbus domestica* L., *Corylus avellana* L. and *Fagus sylvatica* L.

(C) 'Moderate grazing with VMPs' areas—MGVMP—(Catria pastures; 43°30′23.39″N; 12°39′22.39″E). These grasslands are used by cows and horses and have a historical grazing tradition. The farmers there highlighted that VMPs have long been used and this convention continues to today. The unit of livestock/ha is about 1.5 and there is no sign of overgrazing. These pastures are referred to the association *Brizo mediae-Brometum erecti*, where the most abundant species are *Bromus erectus* Huds., *Briza media* L., *Filipendula vulgaris* Moench, *Cyanus triumfettii* (All.) Dostál ex Á.Löve, *Plantago lanceolata* subsp. *lanceolata* (Mert. & Koch), *Luzula campestris* (L.) DC., *Scorzoneroides cichoriacea* (Ten.) Greuter, *Cynosurus cristatus* L., *Anthoxanthum odoratum* L. and *Carex caryophyllea* Latourr. The tree species are represented by the *Scutellario columnare-Ostryetum carpinifolia* association. The sampling sites are located between 800 and 1,000 m a.s.l.

(D) 'Low grazing with VMPs' areas—LGVMP—(Nerone pastures; 43°32′07.27″N; 12°33′26.13″E). These grasslands are grazed by horses that represent a grazing intensity of about 0.5 units of livestock/ha. These sites have been submitted to the historical and intensive use of VMPs from about the 1990s. Today, VMPs are only given to foals and adult animals with evident parasitic stress. The grass associations of these pastures are *Asperulo purpureae-Brometum erecti* and *Brizo mediae-Brometum erecti*, with the principal species being: *Bromus erectus, Briza media, Filipendula vulgaris, Cyanus triumfettii, Plantago lanceolata* subsp. *lanceolata, Luzula campestris, Scorzoneroides cichoriacea, Cynosurus cristatus, Anthoxanthum odoratum* and *Carex caryophyllea*. The arboreous species are dominated by the *Scutellario columnare-Ostryetum carpinifolia* association. The sampling sites are located between 800 and 1,000 m a.s.l.

The density of wild fauna (i.e. *Capreolus capreolus* (L., 1758) and *Sus scrofa* L., 1758) is very similar among all the studied areas (M Tonelli, pers. obs., 2013).

In the areas with VMPs use, the farmers' interviews (M Tonelli, 2013, unpublished data) highlighted that the VMPs have been use since 1990s until today. The main veterinary formulations that are used are based on Ivermectin and Pyrantel pamoate. The main preventive treatments are administrated in spring and in autumn but the data of application vary between each farmer. Moreover, additional treatments are applied as many times as there are parasitic stress. In the LGVMP areas, VMPs are only given to foals and adult animals with evident parasitic stress, but have a very intense historical use of VMPs.

## Dung beetle trapping

For each treatment, we selected three sampling sites separated by at least 500 m to ensure independence among the replicates. In each site, we placed a $50 \times 50$ m quadrate with four pitfall traps at the corners; two traps were baited with cow dung (about 500 $cm^3$) and two with horse dung (about 500 $cm^3$) to maximize differential species attraction (*Barbero, Palestrini & Rolando, 1999*; *Dormont, Epinat & Lumaret, 2004*; *Dormont et al., 2007*). The dung used for the trapping was collected from organic farming that was VMP free. We filled the pitfall traps with propylene glycol (50%) to preserve the dung beetles we collected. The traps were left active for 48 h in each sampling period. The sampling was repeated about every 15 days from June 2013 to November 2013 and in May and June 2014. We excluded rainy days in order to prevent any interference with the trapping. The total number of traps used was 48, and we collected a total of 528 samples (4 traps $\times$ 3 sampling points $\times$ 4 treatments $\times$ 11 sampling periods). The dung beetles were identified to specific level (see Supplemental Information 2, for more details).

## Sampling completeness

The inventory completeness was evaluated using a sample coverage analysis (*Chao & Jost, 2012*). This is a measure of sample completeness, and reveals the proportion of the total number of individuals in a community that belong to the species represented in the sample. The sample coverage formula uses information about sample size, singletons and doubletons (*Chao & Jost, 2012*). Measurements were taken using iNext v.1.0 (*Hsieh, Ma & Chao, 2013*).

## Alpha diversity

Alfa diversity was calculated using the Hill numbers' family diversity (*MacArthur, 1965*; *Hill, 1973*; *Jost, 2006*; *Jost, 2007*) (see Supplemental Information 2, for more details). In order to characterize the complete species abundance distribution and provide full information about its diversity, we computed the diversity of the orders 0, 1 and 2 for each replication of each treatment for the two factors (grazing intensity and VMP use). We then analyzed these results (each order $q$ separately) using a full factorial generalized linear model in order to evaluate the main effect of the two factors and highlight any interactions. Pairwise comparisons were made using the Tukey posthoc test. The diversity profile was produced with SpadeR (*Chao, Ma & Hsieh, 2015*) and the generalized linear model with the Statistica 7.0 package (*StatSoft, 2004*).
### Dung beetle biomass and abundance

We tested the statistical difference in dung beetle total biomass and abundance using a full factorial multivariate generalized linear model with the Statistica 7.0 package (*StatSoft, 2004*) after log transformation of the dependent variable. Pairwise comparisons were made using the Tukey post-hoc test. The average biomass of each species was calculated using the formula 'Biomass $= 0.010864 \times$ Length$^{3.316}$' suggested by *Lobo (1993)*. Ten individuals of each species (when available) were measured to obtain the average species length (see Supplemental Information 2, for more details). To calculate the total biomass of the dung beetle at each treatment, we multiplied the average biomass of each species by the number of individuals collected and added these numbers together.

### Beta diversity

We analyzed whether grazing intensity and VMP use had any effect on the composition of the dung beetle assemblages. We first calculated an index of multiple community similarity of the two factors (using $q = 0, 1, 2$) among all the replicates. This produced six similarity matrices (3 $q$ order $\times$ 2 factors). Based on these matrices, Non-Metric Multidimensional Scaling (NMDS) were constructed and analyzed using a Permanova test (*Anderson, 2001*) to evaluate the statistical significance of each factor for the composition of the dung beetle assemblages at each $q$ level. We computed the multiple community similarity of each treatment with a multiple-assemblage abundance-based overlap measure $C_{qN}$ (*Chao et al., 2008*) (see Supplemental Information 2, for more detail on $C_{qN}$ measures). Similarity matrices were computed using SpadeR (*Chao, Ma & Hsieh, 2015*). A Permanova test was performed using the Permanova+ add-on for PRIMER v.7 (*Anderson, Gorley & Clarke, 2008*; *Clarke & Gorley, 2015*). Interaction between the factors was also evaluated. A total of 999 unrestricted permutations of raw data were computed. The $P$ values were calculated using the Bonferroni correction in all cases.

### Indicator species

The indicator value method (*Dufrêne & Legendre, 1997*) was computed for each factor to identify the indicator species of a particular treatment. This method is used to quantify the value, as a bioindicator, of a set of taxa. In relation to a given species, it combines the measurement of the degree of specificity (how much the species tends to be abundant in a particular ecological state) with the measurement of the degree of fidelity (how much the species tends to be present inside a determined ecological state) with respect to a given ecological status (*McGeoch, Van Rensburg & Botes, 2002*; *McGeoch & Chown, 1998*; *Dufrêne & Legendre, 1997*). The indicator values range from 0 (no indication) to 100 (perfect indication). Species with significant ($P < 0.05$) IndVal results above 70% were considered to be indicator species for the given treatment. Species with an intermediate IndVal between 45% and 70% were considered to be detector species (*McGeoch, Van Rensburg & Botes, 2002*; *Verdú, Numa & Hernández-Cuba, 2011*). Indicator species are highly characteristic of a particular ecological state (treatment) and may decline rapidly under other ecological conditions up to the point of disappearance. Detector species have a different degree of preference for different ecological states, and relative changes in their

abundance across states may be indicative of the direction in which change is occurring (*McGeoch, Van Rensburg & Botes, 2002*). The analysis was performed using PC-Ord 5 (*McCune & Mefford, 1999*).

## RESULTS

A total of 148,668 individuals belonging to 57 species of dung beetle were collected (38 Aphodiidae, 16 Scarabaeidae, three Geotrupidae). This breaks down into: 122,611 specimens belonging to 42 species for the low grazing treatment (25 Aphodiidae, 15 Scarabaeidae, 2 Geotrupidae); 26,057 individuals belonging to 54 species for the moderate grazing treatment (35 Aphodiidae, 16 Scarabaeidae, three Geotrupidae); 128,616 specimens from 53 species for the VMP-free treatment (35 Aphodiidae, 16 Scarabaeidae, two Geotrupidae); and 20,052 individuals belonging to 41 species for the VMP-use condition (24 Aphodiidae, 14 Scarabaeidae, three Geotrupidae) (Supplemental Information 1).

The sample coverage estimator revealed that our inventories were 99% complete for each treatment (Supplemental Information 1). This indicates that only 1% of the individuals in a community belong to species not represented in our samples. We can thus consider our samples to be complete, and we have utilized empirical data for the diversity analysis and comparisons.

### Alpha diversity

Alpha diversity showed a large decrease in the effective number of species as the $q$ order increased, indicating a high degree of dominance in the studied assemblages. There are significant differences in alpha diversity due to the grazing intensity for all $q$ order ($^0D$: $F_{[1,8]} = 62.227$, $P < 0.0001$; $^1D$: $F_{[1,8]} = 48.602$, $P < 0.0005$; $^2D$: $F_{[1,8]} = 34.131$, $P < 0.0005$), with Moderate grazing that have higher equivalent number of species (post-hoc Tukey test $^0D$: $P < 0.0005$; $^1D$: $P < 0.0005$; $^2D$: $P < 0.001$) (Fig. 1).

For VMP use factor significant difference exist only for $^0D$ ($F_{[1,8]} = 242.23$, $P < 0.00001$) whereas no significant difference exist for $^1D$ ($F_{[1,8]} = 0.062$, $P = 0.81$) and $^2D$ ($F_{[1,8]} = 0.041$, $P = 0.85$). Post-hoc Tukey test show that VMP free areas have more equivalent species that VMP use areas for $^0D$ ($P < 0.0005$) but not for $^1D$ ($P = 0.81$) and $^2D$ ($P = 0.85$). A small significant interaction between the two factors was identified only for $^0D$ ($F_{[1,8]} = 5.5$, $P = 0.047$), with post-hoc Tukey test that show significant difference between all experimental groups, with the MGECO areas having 1.11 equivalent species more than the LGECO sites, 1.34 more than MGVMP areas and 1.86 more than the LGVMP sites. Sites with LGECO had 1.21 equivalent species more than the MGVMP areas and 1.66 more than the LGVMP sites. The areas with MGVMP had 1.38 equivalent species more than LGVMP sites.

No significant interaction between the two factors exists for $^1D$ ($F_{[1,8]} = 1.82$, $P = 0.214$) and $^2D$ ($F_{[1,8]} = 0.86$, $P = 0.381$), with post-hoc Tukey test that showed statistical differences only between MGECO areas and LGVMP and LGECO areas, whereas MGVMP had significantly more equivalent species than those of LGVMP and LGECO areas.

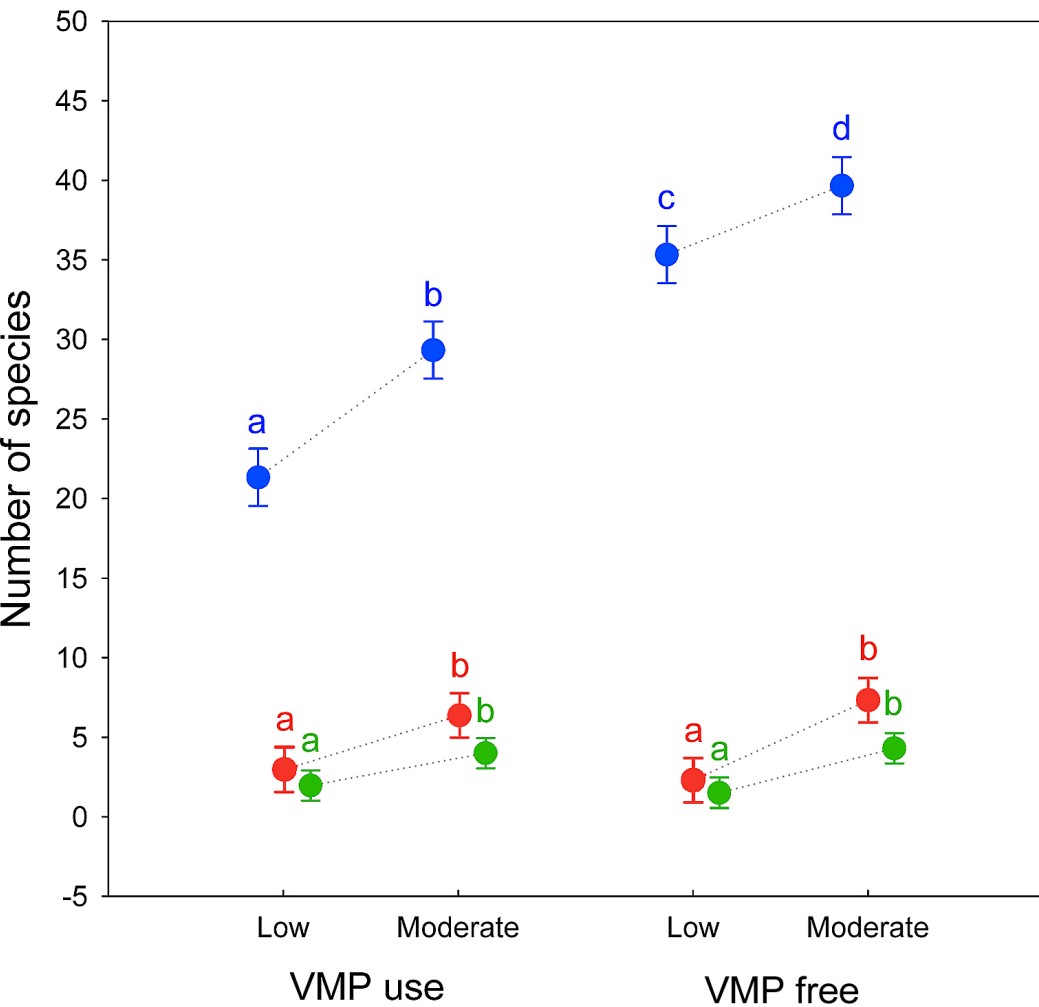

**Figure 1** **Alpha diversity of dung beetles in sub-mountainous landscapes of Central Italy.** Alpha diversity of dung beetles using Hill numbers for both grazing intensity levels (low and moderate) and Veterinary Medical Products use (VMP use and VMP free). $^{0}D$ (blue) correspond to species richness; $^{1}D$ (red) and $^{2}D$ (green) are the alpha diversity indices of $q = 1$ and $q = 2$, respectively. Dots represents mean and bars represent standard errors. Different letters mean significant differences (post-hoc Tukey test $P < 0.05$, with Bonferroni correction).

## Indicator values of species

The IndVal analysis (Table 1) for the grazing intensity factor revealed 10 indicator species: three for the low grazing treatment and seven for the moderate grazing treatment. For the VMP-use factor, 14 indicator species were identified, all with respect to the VMP-free treatment. Two VMP-free indicator species were also indicator species of some treatments for the grazing intensity factor: *Chilothorax conspurcatus* (L., 1758) is an indicator of the VMP-free and low grazing sites, and *Onthophagus taurus* (Schreber, 1759) of the VMP-free and moderate grazing treatments.

**Table 1  Dung beetle indicators of different livestock grazing management techniques.** Numbers represent statistically significant IndVal values ($P < 0.05$).

| Family | Indicator species | LG | MG | ECO | VMP |
|---|---|---|---|---|---|
| Aphodiidae | | | | | |
| | *Aphodius fimetarius* (Linnaeus, 1758) | | | 90.5 | |
| | *Chilothorax conspurcatus* (Linnaeus, 1758) | 93.7 | | 95.9 | |
| | *Melinopterus consputus* (Creutzer, 1799) | 97.3 | | | |
| | *Bodilopsis rufa* (Moll, 1782) | | 97.4 | | |
| | *Calamosternus granarius* (Linnaeus, 1767) | | 83.3 | | |
| | *Labarrus lividus* (Olivier, 1789) | | 76.4 | | |
| | *Melinopterus prodromus* (Brahm, 1790) | | | 99.7 | |
| | *Acanthobodilus immundus* (Creutzer, 1799) | | | 76.1 | |
| | *Nimbus johnsoni* (Baraud, 1976) | | | 79.4 | |
| | *Acrossus luridus* (Fabricius, 1775) | | | 96 | |
| | *Aphodius foetidus* (Herbst, 1783) | | | 83.3 | |
| | *Loraphodius suarius* (Faldermann, 1835) | | | 90.4 | |
| | *Otophorus haemorrhoidalis* (Linnaeus, 1758) | | 100 | | |
| | *Sigorus porcus* (Fabricius, 1792) | 75.6 | | | |
| Scarabaeidae | | | | | |
| | *Onthophagus fracticornis* (Preyssler, 1790) | | 84.3 | | |
| | *Onthophagus ruficapillus* Brullé, 1832 | | | 80.6 | |
| | *Onthophagus taurus* (Schreber, 1759) | | 91.3 | 89.8 | |
| | *Onthophagus coenobita* (Herbst, 1783) | | | 91.3 | |
| | *Onthophagus opacicollis* Reitter, 1892 | | | 100 | |
| | *Bubas bison* (Linnaeus, 1767) | | | 97.2 | |
| | *Copris lunaris* (Linnaeus, 1758) | | 87 | | |
| Geotrupidae | | | | | |
| | *Sericotrupes niger* (Marsham, 1802) | | | 90.5 | |

**Notes.**

LG, low grazing; MG, moderate grazing; ECO, VMP free; VMP, VMP use.

## Biomass and abundance of dung beetles

Significant differences in dung beetle biomass and abundance were obtained for the grazing intensity (*Wilks's lambda* $= 0.138$; $F_{[2,7]} = 21.87$; $P < 0.01$) and use of VMPs factors (*Wilks's lambda* $= 0.17$; $F_{[2,7]} = 17.34$; $P < 0.05$) (Fig. 2). However, no differences were found in their interactions (*Wilks's lambda* $= 0.28$; $F_{[2,7]} = 9.13$; $P = 0.09$). The post-hoc Tukey test showed that the LGECO treatment had a higher dung beetle biomass and abundance than the LGVMP, MGECO and MGVMP treatments, whereas the MGECO treatment had more biomass than the LGVMP treatment.

## Beta diversity

Multiple-assemblage abundance-based similarity measures ($C_qN$) showed a clear aggrupation between sites characterised by both factors studied. For each $q$ level, Non-Metric Multidimensional Scaling (NMDS) plots represent a clear ordination of sites based on grazing activity and VMP use (Fig. 3). The Permanova test showed significant differences in beta diversity for the grazing intensity factor at each $q$ order of similarity matrix (Table 2).

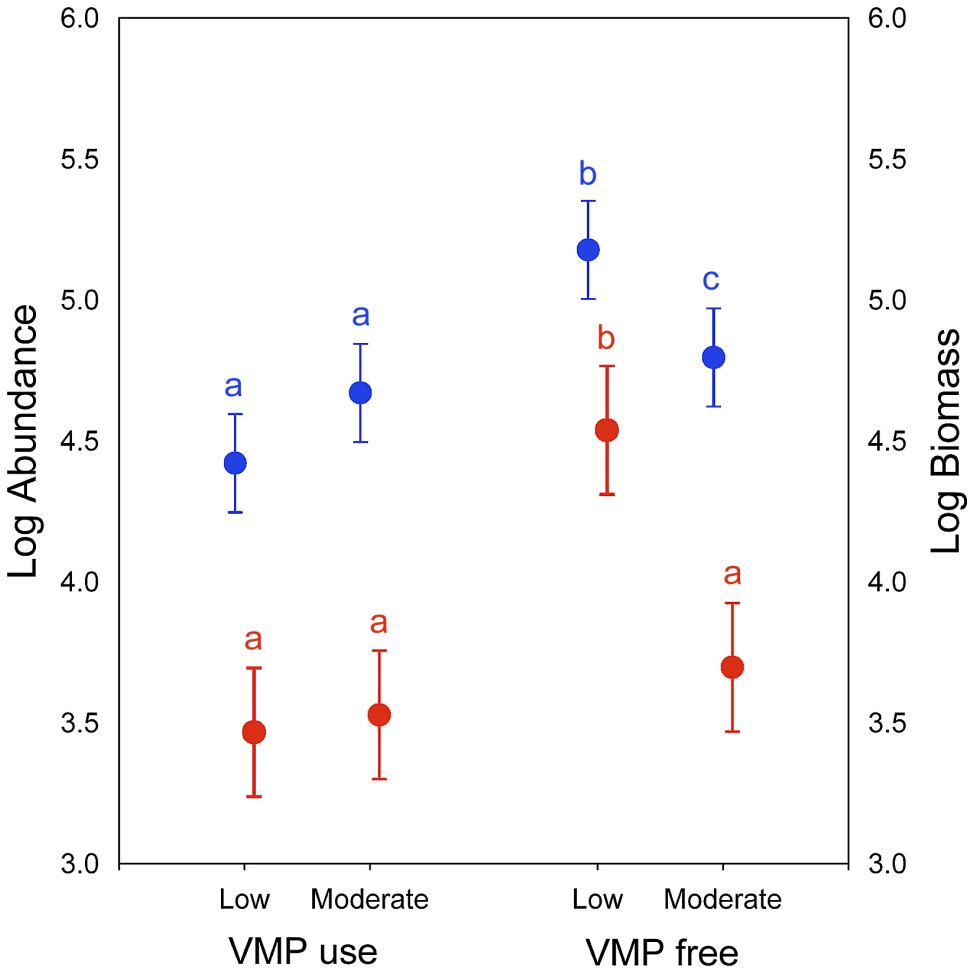

**Figure 2** **Dung beetle biomass and abundance in sub-mountainous landscapes of Central Italy.** Dung beetle biomass (blue) and abundance (red) for different grazing intensity levels (low and moderate) and Veterinary Medical Products use (VMP use and VMP free). Dots represents mean and bars represent standard errors. Different letters mean significant differences (post-hoc Tukey test $P < 0.05$, with Bonferroni correction).

For the VMP-use factor, the Permanova test showed a significant compositional impact only for $q = 0$, whereas it was not significant when species abundance was taken into account, i.e. for $q = 1$ and $q = 2$. Furthermore, the interaction between the two factors was significant only for the similarity matrix of order $q = 0$, but was not significant for $q = 1$ and $q = 2$ (Table 2).

## DISCUSSION

### Grazing intensity effects on dung beetle diversity

Our results support the hypothesis that a low grazing intensity have a negative effect on dung beetle diversity. Total domestic grazing abandonment is a recognised negative factor for dung beetle conservation (*Jay-Robert et al., 2008*; *Verdú, Crespo & Galante, 2000*). However, our results highlighted that even a simple reduction in grazing intensity implies

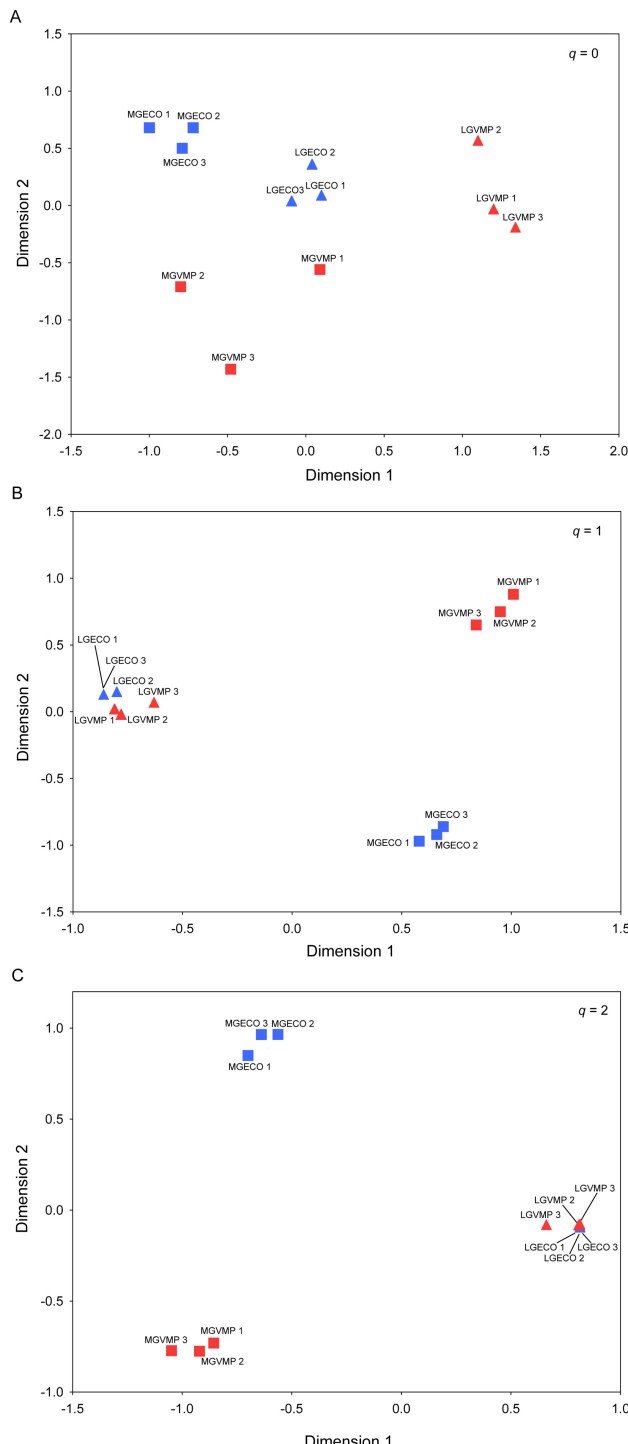

**Figure 3 Beta diversity of dung beetles between sites.** Multiple community similarity using Non-Metric Multidimensional Scaling (NMDS) ordination: (A) Generalised Sørensen index ($C_{0N}$): Average proportion of shared species in each assemblage based on the incidence data; (B) Horn entropy index ($C_{1N}$): proportion of shared species in an assemblage based on abundance data; and (C) Morisita-Horn index ($C_{2N}$): proportion of shared species in an assemblage based on abundance data of the most abundant (dominant) species. A two-dimensional ordination was selected. Each point corresponds to a treatment replication. Squares correspond to moderate grazing areas and triangles to low grazing sites. Sites where VMPs are used are shown in red, whereas sites without any use of VMPs are in blue.

**Table 2 Species compositional similitude among dung beetle assemblages.** The $q$ levels (0, 1 and 2) indicate the value by which multiple community similarity matrices ($C_{q3}$) were calculated.

| Parameter | Source | df | SS | MS | Pseudo-F | P |
|---|---|---|---|---|---|---|
| | GI | 1 | 4960.6 | 4960.6 | 1.0089 | 0.003 |
| | VMP | 1 | 4961.3 | 4961.3 | 1.009 | 0.003 |
| $q = 0$ | GI × VMP | 1 | 4949.2 | 4949.2 | 1.0066 | 0.027 |
| | Residuals | 8 | 39,336 | 4916.9 | | |
| | Total | 11 | 54,207 | | | |
| | GI | 1 | 5118.3 | 5118.3 | 1.043 | 0.003 |
| | VMP | 1 | 4966.5 | 4966.5 | 1.0121 | 0.225 |
| $q = 1$ | GI × VMP | 1 | 4977.3 | 4977.3 | 1.0143 | 0.156 |
| | Residuals | 8 | 39,259 | 4907.4 | | |
| | Total | 11 | 54,321 | | | |
| | GI | 1 | 5234.2 | 5234.2 | 1.0667 | 0.003 |
| | VMP | 1 | 5000.4 | 5000.4 | 1.0191 | 0.213 |
| $q = 2$ | GI × VMP | 1 | 4984.2 | 4984.2 | 1.0158 | 0.258 |
| | Residuals | 8 | 39,255 | 4906.9 | | |
| | Total | 11 | 54,474 | | | |

**Notes.**
GI, the grazing intensity factor; VMP, the VMP-use factor.
P values are calculated using the Bonferroni correction.

negative effects on dung beetle community in areas characterised by a long grazing history, such as the Mediterranean Region. Indeed, the areas with a moderate grazing intensity showed more alpha diversity than the low grazing intensity sites. Thus, our results were consistent with those of other studies in different Mediterranean locations. For example, *Lobo, Hortal & Cabrero-Sañudo (2006)*, in Spain, showed that the quantity of dung in a radius of 2 km and the presence of a flock are key factors in determining the local variation in dung beetle species richness and abundance. In Southern France, *Lumaret, Kadiri & Bertrand (1992)* explained that an increase of 260% in fresh dung availability, five years after a change of pasture management (from sheep to cows), caused an increase in species richness from 38 to 42 species. In Italy, *Carpaneto, Mazziotta & Piattella (2005)* showed that after 13 years, the abandonment of the sheep grazing system in the Rome urban area led to a loss of 53% of the dung beetle species, especially those with a large body size.

Furthermore, the decrease in the number of indicator species observed in function of the decrease in grazing intensity supported our hypothesis. We encountered seven and three species with significant IndVal values for the moderate and low grazing areas, respectively. This fact suggest that a reduced quantity of a trophic resource can favour a limited number of species. Moreover, it is interesting to note that in the moderate grazing sites studied, three of the seven indicator species are paracoprids and, among them, one, *Copris lunaris* (L., 1758), is a large species. During breeding, *Copris lunaris* may bury about 100–165 g of dung (*Klemperer, 1982*; *Martín-Piera & López-Colón, 2000*). Thus, there is a positive relationship between body size and dung mass burial (*Doube, Giller & Moola, 1988*; *Larsen, Williams & Kremen, 2005*; *Slade et al., 2007*), which supports the idea that large paracoprid

dung beetle, as *C. lunaris*, can only to maintain well established populations if the trophic resource is relatively abundant.

Our results on the grazing intensity factor can be explained by the species–energy relationship (*Gaston, 2000*; *Wright, 1983*; *Hawkins et al., 2003*), i.e., the lower the level of (trophic) energy available, the smaller the number of species that an area can support (*Evans, Warren & Gaston, 2005*). For example, *Tshikae, Davis & Scholtz (2013)* explicitly tested the species–energy relationship for dung beetles across an arid and trophic resource gradient in Botswana. Their results showed that the species richness, diversity and biomass of the dung beetle diminish with a decrease in available (trophic) energy.

However, it is interesting to note that the low grazing areas studied showed greater biomass and abundance. This may be explained by the dominance of two species, *Melinopterus consputus* (Creutzer, 1799) and *Onthophagus medius* (Kugelann, 1792). Both species may alter the diversity pattern of this treatment by means of a competitive exclusion (*Hardin, 1960*). The low quantity of the trophic resource available in this site perturbed the dung beetle community, favouring generalist r-strategic species (such as *M. consputus*) and highly competitive species such as small tunnellers (e.g., *O. medius*) (*Horgan & Fuentes, 2005*). The low grazed sites studied, in fact, showed more biomass but fewer species than the moderately grazed areas. The same results were reported in the Rome urban area (Italy) by *Carpaneto, Mazziotta & Piattella (2005)*, who found a decrease in the number of species and a rise in total biomass, with the dominance of one species of Aphodinae with the same explosive reproductive strategy (i.e., *Nimbus johnsoni* (Baraud, 1976)).

In terms of species composition of assemblages, beta diversity was strongly influenced by the quantity of the trophic resource at all *q* levels (Table 2); rare and abundant species were compositionally different between the assemblages obtained in the different grazing intensity treatments. These results implicate that grazing extensification lead to a change in dung beetle composition favouring more opportunistic species. This explanation is corroborated by the presence of three indicators species (*Melinopterus consputus*, *Chilothorax conspurcatus* and *Sigorus porcus*) characteristics of the low grazing areas that share an opportunistic behaviour. *Melinopterus consputus* and *C. conspurcatus* are dependent on the dung only during adult stage, whereas during larvae phase are saprophagous, mainly (JR Verdú, pers. obs., 2004); *Sigorus porcus* have a strong attitude to kleptoparasitism during both adult and larval stages (*Dellacasa & Dellacasa, 2006*).

Thus, dung beetles are strongly dependent to dung during their life cycle and our data support the hypothesis that even a simple reduction of its availability may have negative effects on the community. Less trophic resource availability lead to a compositional and structural impoverishment of the community with a loss of large body sizes dung beetles in favour of more opportunistic ones. Then, the fact that Mediterranean pastures suffer a continuous process of extensification, can be a factor of concern for the dung beetle conservation.

## VMPs use effects on dung beetle community

Our results supported the hypothesis that the historical use of VMP substances have a negative effect on dung beetle diversity. The negative effect of VMP substances was relevant to all community parameters measured, such as alpha diversity, biomass, abundance, presence of indicator species and beta diversity. It has been documented that VMP-use shows a variety of lethal and sub-lethal effects on non-target fauna depending on the molecule, doses, mode of administration, environmental factors and insect species in question (*Lumaret & Errouissi, 2002*; *Lumaret et al., 2012*; *Wall & Beynon, 2012*; *Jacobs & Scholtz, 2015*). Many essays show that VMPs negatively affect larval and adult survival of dung beetles, as well as some physiological processes such as reproductive, sensorial and locomotor capacities showing even negative repercussions in the dung decomposition (*Wall & Strong, 1987*; *Lumaret et al., 1993*; *Wardhaugh, Longstaff & Morton, 2001*; *Verdú et al., 2015*).

Here, we document that VMP-use sites studied showed significantly fewer species and a reduced biomass compared to the VMP-free sites. Our results agree with other studies that have explored the impact of VMPs in the field. For example, in southern Ireland, *Hutton & Giller (2003)* observed a lower number of species and a reduced abundance of dung beetles in intensive and rough grazing farms compared to organic farms. In South Africa, *Krüger & Scholtz (1998)* also showed that, under drought conditions, treatment with ivermectin led to a loss of dung beetle species. *Beynon et al. (2012b)* showed a reduction in dung beetle abundance and biomass in dung treated with ivermectin in the UK.

Unlike some studies (*Krüger & Scholtz, 1998*; *Basto-Estrella et al., 2014*; *Hutton & Giller, 2003*), we did not find a significant difference in $^1D$ ('common species number') and $^2D$ ('dominant species number') for the VMP-use factor. Give that macrocyclic lactones as ivermectin acting on a family of ligand-gated chloride channels gated by glutamate, which is shared by all Ecdysozoan (*Geary & Moreno, 2012*; *Puniamoorthy et al., 2014*), all dung beetles species should be sensible to ivermectin toxicity. Thus, the consequences on the assemblage structure may be differential based on the abundances of each species in each assemblage. Our data showed that less common species are first in disappearing in sites characterized by VMPs use, which explains the significant reduction in the number of species observed in these sites. At $q = 1$ and $q = 2$, however, differences are not observed between both treatments, so the reduction of the populations of the most common and dominant species took place of equitable way, which maintains similar measures of community structure ($^1D$ and $^2D$).

Accordingly to alpha diversity results, beta diversity was influenced by the use of VMPs only for $q = 0$. This means that the two assemblages are different in terms of 'rare' species, whereas the more common and dominant species are not significantly different.

Our IndVal results showed how the VMP-use treatments have no indicator species. This means that no species were favoured by the use of these veterinary substances. In other words, the use of VMPs could affect all species and, apparently, no species could be resistant to VMP toxicity. These results agree with the explained above about diversity measures. In contrast, the VMP-free treatment had 14 indicator species.

Our results are congruent with those of *Puniamoorthy et al. (2014)*, which show that ivermectin sensitivity is an ancient trait affecting potentially all Ecdysozoan (moulting animals) species. This corroborates the hypothesis that the use of VMPs may have a ubiquitous, negative effect on dung beetle fauna. The fact that no species were found to be indicator species in the areas with VMP-use could be due to the irrational use of these substances throughout the year.

## Grazing intensity and VMPs interactions

Interesting results were highlighted by the interactions between the two factors. Contrary to our hypothesis, the interactions terms were less significant, or no more significant, than the main effects (each factor separately) for alpha diversity, biomass and species composition. This could be explained if we consider that both factors affect different forms of each species. In other words, the decrease in dung availability affects several bigger species more than the very small species, but this does not imply that the former are more susceptible to injury caused by the ingestion of dung contaminated with VMPs. Another explanation can be found in the halving of the sample size during the interaction analyses. This means that interactions between the two factors may have antagonistic effects on dung beetle assemblages, but more studies with greater sample size are needed on this issue.

## CONCLUSIONS

The present analysis highlighted that the moderate grazing VMP-free treatment seems to be the more appropriate management system for maintaining a higher number of dung beetle species, as well as greater diversity and biomass. These results corroborated the notion that, in a Mediterranean context with a long history of grazing, traditional management techniques with a moderate grazing intensity have a positive effect on dung beetle diversity (*Verdú, Crespo & Galante, 2000*). Furthermore, our results corroborated the hypothesis that both factors—low grazing intensity and VMP-use—have negative effects on dung beetle communities. Even a simple grazing intensity extensification may have negative impact on dung beetle, that is reflected in the compositional and structural impoverishment of the community. Our study strengthens the results about the environmental risk assessment made by *Liebig et al. (2010)* that concluded that the ivermectin use have an "unacceptable risk" for dung beetle fauna.

The results could have an application for sustainable farmland management, highlighting that an incorrect grazing management of the pastures could be a strong effect on dung beetle community (e.g., number of species, biomass, composition), and so in the correct function of ecosystem processes performed by dung beetle as nutrient cycles, vegetation development, secondary seed dispersal, and parasite control (*Nichols et al., 2008*; *Nervo et al., 2014*; *Beynon et al., 2012a*; *Larsen, Williams & Kremen, 2005*). Then, the loss of dung beetle biodiversity can have a negative impact on various ecosystem processes (*Nichols et al., 2008*), with harmful effects on pastures.

Finally, we suggest that organic farming with a moderate grazing intensity could have a positive effect on dung beetle conservation. This farming management approach may contribute to this by avoiding pasture abandonment, conferring an economic stimulus

(*Willer & Lernoud, 2016*) and controlling for the excessive use of VMPs (*Hutton & Giller, 2003*). Further studies in different biogeographical and bioclimatic regions are, however, needed to assess the impact of the long-term use of VMPs on dung beetles.

## ACKNOWLEDGEMENTS

Sally-Ann Ross checked the English version of the manuscript. We would like to thank Rodrigo F. Braga and one anonimous reviewer for their many insightful comments and suggestions. We are grateful to Marco Dellacasa for his help about the identification of some cryptic Aphodiinae species.

### Funding

Financial support was partially provided by Project CGL2015-68207-R of the Secretaría de Estado de Investigación, Desarrollo e Innovación of the Ministerio de Economía y Competitividad of Spain. Mattia Tonelli benefited for an Italian ministerial PhD scholarship. The funders had no role in study design, data collection and analysis, decision to publish, or preparation of the manuscript.

### Grant Disclosures

The following grant information was disclosed by the authors:
Secretaría de Estado de Investigación, Desarrollo e Innovación of the Ministerio de Economía y Competitividad of Spain: CGL2015-68207-R.

### Competing Interests

The authors declare there are no competing interest.

### Author Contributions

- Mattia Tonelli and José R. Verdú conceived and designed the experiments, performed the experiments, analyzed the data, contributed reagents/materials/analysis tools, wrote the paper, prepared figures and/or tables, reviewed drafts of the paper.
- Mario E. Zunino conceived and designed the experiments, performed the experiments, contributed reagents/materials/analysis tools, wrote the paper, reviewed drafts of the paper.

### Data Availability

The raw data has been supplied as a Supplemental File.

### Supplemental Information

Supplemental information for this article can be found online at http://dx.doi.org/10.7717/peerj.2780#supplemental-information.

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
