# Peer review of "Effects of grazing intensity and the use of veterinary medical products on dung beetle biodiversity in the sub-mountainous landscape of Central Italy"

_PeerJ, doi:10.7717/peerj.2780_

## Round 0.1 · original submission · Major Revisions

· Academic Editor

Major Revisions

Both reviewers have raised substantive issues with that manuscript which need to be addressed. I also read the manuscript and agree with the issues raised.

Reviewer 1 ·

Basic reporting

1. Basic reporting. The manuscript is well written, clear, and the subject about the effects of veterinary medical products on benefical fauna in managed grasslands is a relevant current topic.
Specific comments on this section.
L56. Reference of Spector 2006 is not in the references section.
L86. Reference of Wardhaugh et al. 1988. Actors should not use “et al.” according with the references section. Must be Wardhaugh & Rodriguez-Menendez 1988.
L184-188. Please change the ampersand for “and” in the references of Chao and Jost 2012.
L191. Reference of McArthur 1965 is not in the references section.
L281-284. Please, provide more details on this result, provide values and consider the possibility of include a figure or a table with the resulting values of the analysis of sampling effort.
L315-321. I assumed that Figure 2 belongs to the results described in this paragraph, but there is any indication of this in the text. Please refer the figure 2 in the text.
L500. Reference of Colwell 2013 is not cited in the text.
L559. The reference of Hsieh et al. 2013 is not in accordance with the alphabetical order of the list.
Figure1. Figures must be auto explanatory. The figure must specify what are the bars in the dots. I suspect that are standard errors but authors should declare it in the figure legend. Additionally authors must specify what is VMP and refer properly to low-modarete grazing, since the figure do not explain this clearly.
Figure2. The same comment of figure one. Additionally, figure legend remarks the statistically differences are marked with different letters, but the figure does not include any letter.
Table1. I suggest to add in the legend: “The numbers represent significant IndVal values (p < 0.05).”

Experimental design

2. Experimental design. The experimental design is well explained, nonetheless some aspects are of concern, because their posible effect is directly over the main results of this manuscript.
2.1. I have a clear concern on the pseudo-replication of the samples in the design, since there are four ranches and tree samples in each ranch, but each ranch represent a hole treatment. My question on this is. How representative of the treatments are each of this ranches? It makes sense in terms of the number of beetles collected, but i have doubts on how a single ranch is going to represent an entire condition as might be the use os VMPs or the grazing intensity. It is very important that authors explain clearly the scope and implications of this problem in the results of the study.
2.2. One of the important things that I observed in the results is the certain effect that moderate grazing-VMPs free samples has on beetle diversity, but curiously, is the lowest site (altitude), so authors need to explain how difference in altitud is avoiding the probable bias that this represent.
2.3. There should be a paragraph or table were the authors describe which VMPs are used in the sites, how many time (at least approximately) the VMPs have been used in this sites, in general terms authors must provide details about this issue, since is one of the principal factors evaluated.
2.4. I suggest to describe if there is a seasonal abundance of the dung beetle communities in the studied area, since many scientific papers describe seasonality in this organisms. So it would be very helpful to know which are the months of greatest abundance in the area.
2.5. The design of the study avoid the possible effect of the type of bait in the traps. This issue must be addressed because the statistical analysis do not consider this as a factor, and the variation in abundance, biomass and/or diversity could be affected by the type of bait used for trapping.
2.6. In L223 the text mention “…average weight of each species…” How average weight was measured? The methodology section explain how biomass and abundance was calculated but there is any description on the weight. Please describe this procedures and explain how this information was used, since there is no explanation of its analysis.
2.7. In L265-266 the authors state the values for IndVal to consider a significant indicator species. Please provide reference that support this quantities to consider this values a good measure of the indicator value of a species.
2.8. My main concern with the analysis of the data is the use of the Hill number’s family diversity. I doubt of this approach, because each of the numbers are in different scales, so that could hinder the analysis with the generalized linear model proposed by the authors. For example, in the results the authors describe that they found differences between each factors but they did not observed interactions between the factors. This is probably because the order q1 will never be greater than order q0, mostly because the index will be estimating a greater number of species than those that in fact were observed. In other words, the orders q1 and q2 will never be greater than order q0 so interaction between factors is unreliable. Authors must explain properly the support for the use of this analysis approach. The same for the PERMANOVA analysis.

Validity of the findings

The validity of the findings depend on the answers provided on the questioning of the design.

Additional comments

The manuscript is interesting and very fluid to read. The arguments are well written and the results are dependent on the design of the study, so until questions about the design are properly solved, I can not take into account the validity of results.

·

Basic reporting

"No Comments".

Experimental design

"No Comments".

Validity of the findings

"No Comments".

Additional comments

The work “Effects of grazing intensity and the use of veterinary medical products on dung beetle biodiversity in the sub-mountainous landscape of Central Italy” presents scientific merit and shows results important, mainly, for being about the reader’s understanding concerning ecologically important groups and bioindicators in areas of human production. The conservation future must be seen in those areas, since they tend to dominate the landscape of our planet. Understanding the effect of the veterinary medical products in non-target groups is also very important to the conservation and maintenance of natural and man-modified.
The sampling and the data obtained were capable of responding the proposed objectives.
My greatest concern is in the discussion and in the bibliographical review conducted. About 25 % of your references are of the last five years. A more updated review about the subject could improve the discussion. Biological implications of your results would ennoble the discussion which is very superficial.
A few suggestions of references:
Braga, R.F., Korasaki, V., Audino, L.D. et al. Are Dung Beetles Driving Dung-Fly Abundance in Traditional Agricultural Areas in the Amazon?Ecosystems (2012) 15: 1173. doi:10.1007/s10021-012-9576-5
Liebig M, Fernandez ÁA, Blübaum-Gronau E, Boxall A, Brinke M, Carbonell G. Environmental risk assessment of ivermectin: A case study. Integr Environ Assess Manag. 2010; 6: 567–587. doi:10.1002/ieam.96
Rodríguez-Vivas RI, Pérez-Cogollo LC, Rosado-Aguilar JA, Ojeda-Chi MM, Trinidad-Martinez I, Miller RJ, et al. Rhipicephalus (Boophilus) microplus resistant to acaricides and ivermectin in cattle farms of Mexico. Rev Bras Parasitol Vet. 2014; 23: 113–22. doi:10.1590/S1984-29612014044
Osei-Atweneboana MY, Awadzi K, Attah SK, Boakye DA, Gyapong JO, Prichard RK. Phenotypic Evidence of Emerging Ivermectin Resistance in Onchocerca volvulus. PLoS Negl Trop Dis. 2011; 5(3): e998. doi:10.1371/journal.pntd.0000998
Scholtz CH, Jacobs C. A review on the effect of macrocyclic lactones on dungdwelling insects : Toxicity of macrocyclic lactones to dung beetles. Onderstepoort J Vet. 2010; 1–8. doi:10.4102/ojvr.v82i1.858

Ascher KRS. Nonconventional insecticidal effects of pesticides available from the Neem tree, Azadirachta indica. Arch Insect Biochem Physiol. 1993; 22: 433–449. doi:10.1002/arch.940220311
Dadour IR, Cook DF, Neesam C. Dispersal of dung containing ivermectin in the field by Onthophagus taurus (Coleoptera: Scarabaeidae). Bull Entomol Res. 1999; 89: 119–123. doi:10.1017/S000748539900019X
Alves S, Serrão J. Effect of ivermectin on the life cycle and larval fat body of 62 Culex quinquefasciatus. Braz Arch Biol Techn. 2004; 47: 433–439. doi:10.1590/S1516-89132004000300014
Geary, T. G. & Moreno, Y. Macrocyclic Lactone Anthelmintics: Spectrum of Activity and Mechanism of Action. Curr. Pharm. Biotechnol. 13, 866–872 (2012).
Lumaret, J. P., Errouissi, F., Floate, K., Römbke, J. & Wardhaugh, K.A Review on the Toxicity and Non-Target Effects of Macrocyclic Lactones in Terrestrial and Aquatic Environments. Curr. Pharm. Biotechnol. 13, 1004–1060 (2012).

Some commentaries and suggestions to the improvement of the text can be seen below.
Page 126. First citation of the scientific name must be accompanied of both author and date. All the first citations of scientific names must be revised in the text.

The vegetation structure could not have influenced the results? Since they are different among the areas? The species diversity cannot be so important to beetles, but the vegetation structure can. Comment about it.

I noticed that there is an altitudinal gradient. Cannot it have interference? Even small altitudinal variations can influence the beetle community. (See Nunes CA, Braga RF, Figueira JEC, Neves FdS, Fernandes GW (2016) Dung Beetles along a Tropical Altitudinal Gradient: Environmental Filtering on Taxonomic and Functional Diversity. PLoS ONE 11(6): e0157442. doi:10.1371/journal.pone.0157442).

In table 1, put the family of the species or subfamily.

Place a figure with the localization (map) of the collecting sites.

The description of the surveys is a bit long. Present here only essential information of the surveys, a more complete description could be presented in a supplementary material.

Line 270. I would like to see that description with family or subfamily.

Supplementary Material S1 – I would like to see in that table a family/subfamily classification.

I suggest that the topics presented in the results are according to the objectives not with the surveys utilized.

Line 315. Here should figure 2 not be indexed?

Are the letters in the plot of figure 2 not lacking 2? “Different letters mean significant differences (post-hoc Tukey test P < 0.05)”.

Discussion. We could try to follow the topics of the results. That would make the reader’s understanding easier.
I see that your discussion analyze little your results. I do not seed the biological implications of your data. I believe that the separation into topics based on your objectives will help improve the discussion. The discussion deserves more attention.

Line 393. But your areas showed no differences in the abundance between areas with and without ivermectin. If that were true, should you not have found that difference? That contradicts your results.

You have about 7 references of the last 5 years in your discussion; I think an increased discussion would benefit your work.

---

## Round 0.2 · accepted · Accept

· Academic Editor

Accept

I am happy with the changes you have made to the manuscript in respect to the reviewers comments.

Reviewer 1 ·

Basic reporting

No more comments, all the previous requests and comments were answered by authors

Experimental design

No more comments, all the previous requests and comments were answered by authors

Validity of the findings

After logical and referenced answers stated in the authors' response letter, my opinion is that the findings of this study should not be generalized. The findings are interesting and should be corroborated in a broader study. I still think that a more replicated design should be performed. Nonetheless, the results are valid and well explained.

Additional comments

I encourage the authors to improve the design of future studies by a more replicated strategy. I am not saying that the current study has a wrong design, but the better the design, the better the conclusions that can be found. Particularly in this conservation and management topic.
It is urgently needed that field studies provide more evidence of strategies for a better cattle and grassland management to maintain ecological services provided by dung beetle communities.